# Prostate cancer in Spain: A retrospective database analysis of hospital incidence and the direct medical costs

Josep Darbà[1]*, Meritxell Ascanio[2]

1 Department of Economics, Universitat de Barcelona, Barcelona, Spain, 2 BCN Health Economics & Outcomes Research S.L., Barcelona, Spain

* darba@ub.edu

## Abstract

### Introduction

The goal of this study is to determine the medical costs, comorbidity profile, and health care resources use of patients diagnosed with prostate cancer who have been treated in Spanish hospitals.

### Methods

The admission records of the patients diagnosed with prostate cancer used in the study were registered between January 2016 and December 2020. These records have been collected from a Spanish hospital discharge database and have been evaluated in a retrospective multicenter analysis.

### Results

8218 patients from the database met the criteria and were thus analyzed. The median aged of the diagnosed patients was 71.68 years. The median Charlson comorbidity index (CCI) score was 4, and the updated median CCI was 3. Hypertension was diagnosed in the 49.76% of the individuals, 37.03% had chronic obstructive pulmonary disease and 34.51% had hyperlipidaemia. The mortality rate was 9.30%. The most common medical procedure was prostate resection with percutaneous endoscopic approach (31.18%). The mean annual cost per admission was 5212.98€ €.

### Conclusions

Technologies, such as the prostate-specific antigen (PSA) testing for screening has helped in the diagnosis in the past decades, enhancing a decrease in the mortality rate of the patients throughout the years.

**Data Availability Statement:** Data are available upon request to the Spanish Ministry of Health for any interested researcher, yet data sharing is restricted due to legal stipulations. Requests must be addressed to the Unit of Health Care Information

and Statistics (Spanish Institute of Health Information) https://www.sanidad.gob.es/estadEstudios/sanidadDatos/home.htm.

**Funding:** The author(s) received no specific funding for this work.

**Competing interests:** The authors have declared that no competing interests exist.

## Introduction

Prostate cancer is the malignant neoplasm of the prostate, the vast majority of which are of epithelial origin and differentiation and are classified as carcinomas [1]. It is the most frequent diagnosed non-cutaneous malignant neoplasm in men in Europe and the second most frequently diagnosed malignancy in the world [2–4]. Worldwide 1,111,000 cases were diagnosed in 2012, where 417,000 cases were diagnosed in Europe [2]. According with the International Agency for Research on Cancer, prostate cancer is the most common type of cancer diagnosed in men worldwide, and was the second most common cancer in 2012, accounting with a 15% of the total cases [5]. In Spain, prostate cancer is the third leading cause of cancer death in men, following lung and colorectal cancer [5].

Over the last 20 years, the incidence of prostate cancer has increased in Europe due to the introduction in the late 1980s of the use of the prostate specific antigen (PSA) test for screening, which allows the detection of more cases [6]. The use of PSA alongside the Gleason score has been used to classify disease risk and help guide treatment decisions in recent years [7].

In Spain, the incidence rate was 20.25 per 100,000 men in 2010, lower than in other EU countries [8]. Epidemiological and biological data increasingly show that risk, aggressiveness, and prognosis vary by race, ethnicity and geography [9].

This study will aim to analyse and describe the in-hospital incidence and identify any factor that may play a role in the prostate cancer mortality. In addition, the study will assess any time trends and quantify the associated medical costs.

## Methods

### Study design

Hospitals admissions records of patients with prostate cancer in Spain between 1 January 2016 to 31 December 2020 were analyzed in a retrospective multicenter study. The data collected come from a database that codifies data at the hospital level by using the 10th version of the International Statistical Classification of Diseases and Related Health Problems (ICD-10). Inpatient admissions records were collected from a Spanish National discharge database.

The database contains information of 90% of hospitals in Spain and includes data from all regions. Errors and unreliable data are eliminated during the data validation, which is carried out internally and subjected to regular audits. The data codification, evaluation and confidentiality are the responsibility of all the centers.

### Data extraction

The ICD-10 codes C61 and R97.21 were used to identify patients in the database. All parameters regarding healthcare centers and medical histories were re-coded to maintain anonymity in accordance with the principles of Good Clinical Practice and the Declaration of Helsinki. The analysis did not need the involvement of human participants and there was no access to identifying data. In the research context, the Spanish legislation does not require patient consent and ethics committee approval in studies with anonymized data.

### Study variables

The study variables that have been evaluated in the study include the patient's age, national region, type of admission, discharge type (including death), funding scheme, service, intensive care unit, length of hospital days, primary diagnosis, 19 secondary diagnoses, medical procedures, readmission, cancer morphology, mortality risk and total admission cost.

## Data analysis

Patients with prostate cancer were selected by the primary diagnosis code. They were organized by age into five different groups (below 18, between 18 and 45, between 45 and 65, between 66 and 85, and over 85). To assess the characteristics of each patient, the first admission registered per patient was used. In the subsequent analysis all admissions records for admission details and the direct medical costs. The ratio between admission and hospitalization rate per 10,000 persons was used as the hospital incidence.

The direct medical costs were assigned to each hospital admission according to the average admission costs and standardized medical procedures, previously identified by the Spanish Ministry of health. Total expenses include all costs regarding medical examinations, medications, procedures, medications, diet, medical equipment, costs associated to personnel, and resources. Regarding costs information, in Spain, healthcare is free of charge, so the costs provided will be those paid by the National Health System.

To test normality the Kolmogorov-Smirnov test was used in all data. Frequencies and percentages are shown for dichotomous variables and mean, or median were assessed for continuous variables. The Kruskal-Wallis test as a one-way analysis of or the Mann-Whitney U test as a two-tailed non-parametric independent t-test were used as appropriate and two-sample Z tests were used to distinguish in sample proportions. The Jonckheere-Terpstra trend test was used to evaluate trends in incidence and cost. A p-value below 0.05 was considered to be statistically significant.

StataSE 12 for Windows (StataCorp LP. 2011. Stata Statistical Software: Release 12. College Station, TX, USA) was utilized to carry statistical analysis.

## Results

A total of 8882 admissions, which corresponded to 8217 patients, were included during the study period. Median age was 71.68 years old, with 66.32% of the cases registered between the age range of 66 and 85 years old, 24.95% were between 45 and 65 years old, and 8.64% of the cases were over 85 years old. Only 7 individuals were in the age range of 18 and 44 years old. Median length of hospital stays was 7 days (Table 1).

The main diagnosis of all individuals was malignant neoplasia of the prostate. The most common secondary diagnosis was hypertension with a 49.76% of the individuals, followed up

**Table 1. Patient baseline characteristics.**

|  | Total |
|---|---|
| **Admissions, N** | 8882 |
| **Patients, N** | 8217 |
| 0–17 years, N (%) | 0 (0) |
| 18–44 years, N (%) | 7 (0.09) |
| 45–65 years, N (%) | 2050 (24.95) |
| 66–85 years, N (%) | 5450 (66.33) |
| >85 years, N (%) | 710 (8.64) |
| **Average age** | 71.68 |
| 2016 | 72.08 |
| 2017 | 72.04 |
| 2018 | 71.15 |
| 2019 | 71.62 |
| 2020 | 71.52 |

by chronic obstructive pulmonary disease (37.03%) and hyperlipidaemia (34.51%). A 23.45% of the cases had a personal history of nicotine dependence and 22.28% had type 2 diabetes mellitus (Table 2).

Although some of the patient underwent more than one procedure, there was also some of them that underwent any procedure, so the mean number of procedures underwent per patient is 2.4. The most common medical procedure was prostate resection with a

**Table 2. Patient comorbidity profile.**

|  | Index admission | p-value [a] |
|---|---|---|
| **Admissions, N** | 8882 | - |
| **Mean CCI** | 4.60 | < 0.0001 |
| **Median CCI** | 4 | - |
| CCI 0, N (%) | 0 (0) | < 0.0001 |
| CCI 1, N (%) | 1 (0. 01) | < 0.0001 |
| CCI 2, N (%) | 874 (10.64) | < 0.0001 |
| CCI 3, N (%) | 2841 (34.57) | < 0.0001 |
| CCI 4, N (%) | 1600 (19.47) | < 0.0001 |
| CCI 5, N (%) | 625 (7.61) | < 0.0001 |
| CCI > 5, N (%) | 2277 (27.71) | < 0.0001 |
| **Mean updated CCI** | 4.22 | < 0.0001 |
| **Median updated CCI** | 3 | - |
| CCI 0, N (%) | 0 (0) | < 0.0001 |
| CCI 1, N (%) | 1 (0.01) | < 0.0001 |
| CCI 2, N (%) | 1283 (15.61) | < 0.0001 |
| CCI 3, N (%) | 3201 (38.96) | < 0.0001 |
| CCI 4, N (%) | 1284 (15.63) | < 0.0001 |
| CCI 5, N (%) | 379 (4.61) | < 0.0001 |
| CCI > 5, N (%) | 2070 (25.19) | < 0.0001 |
| **Secondary diagnoses, >10%** |  |  |
| Malignant neoplasm of prostate | 8216 (99.99) | < 0.0001 |
| Essential (primary) hypertension | 4089 (49.76) | < 0.0001 |
| Chronic obstructive pulmonary disease | 3043 (37.03) | < 0.0001 |
| Hyperlipidemia | 2836 (34.51) | < 0.0001 |
| Personal history of nicotine dependence | 1927 (23.45) | < 0.0001 |
| Uncomplicated type 2 diabetes mellitus | 1831 (22.28) | < 0.0001 |
| Old myocardial infarction | 1573 (19.14) | < 0.0001 |
| Secondary malignant neoplasm of bone | 1317 (16.03) | < 0.0001 |
| Chronic ischemic heart disease | 1262 (15.36) | < 0.0001 |
| Presence of coronary angioplasty, grafts and prostheses | 1214 (14.77) | < 0.0001 |
| Unspecified asthma, without complications | 1043 (12.69) | < 0.0001 |
| Long-term (current) use of aspirin | 968 (11.78) | < 0.0001 |
| **Mortality rate, %** |  |  |
| **2016** | 10.45 | - |
| **2017** | 9.84 | - |
| **2018** | 9.75 | - |
| **2019** | 8.72 | - |
| **2020** | 7.69 | - |

[a] index admission; CCI, Charlson's Comorbidity Index

**Table 3. Medical procedures.**

| | Index admission | p-value [a] |
|---|---|---|
| **Medical procedures, >10%** | | |
| Prostate resection, percutaneous endoscopic approach | 2562 (31.18) | < 0.0001 |
| Seminal vesicle resection, bilateral, percutaneous endoscopic approach | 2093 (25.47) | < 0.0001 |
| Prostate resection, open approach | 1087 (13.23) | < 0.0001 |
| Bladder drainage, with drainage device, natural or artificial orifice approach | 851 (10.36) | < 0.0001 |

[a] index admission

percutaneous endoscopic approach (31.18%) and seminal vesical resection with bilateral percutaneous endoscopic approach (25.47%) (Table 3).

Regarding the Charlson's Comorbidity Index (CCI), a 34.57% of the patients had a CCI score of 3, while over 27% had a CCI score over 5. Furthermore, regarding the updated CCI there was a score of 3 for 38.96% of the patients (Table 1).

The incidence was 30.72 per 10,000 habitants in 2016, while in 2020 it decreased to 16.14 (p<0.0001) (Fig 1). The mortality rate in 2016 was 10.45%, while it was also reduced in 2020 to a 7.69% (Fig 2, Table 2). The mean annual cost per each hospital admission was 5212.98€ (p<0.0001), meanwhile the total cost was 42.87 million €, with a mean annual cost of 8.57 million € (Fig 3).

## Discussion

This retrospective study assessed the incidence and costs of prostate cancer in Spain. This study found an incidence of 20.30 per 10,000 persons over the study period. The incidence rate decreased over the years. The average age in the study was 71.68 years of age. As well, other studies show the main population diagnosed with prostate cancer to be over 70 [9–11].

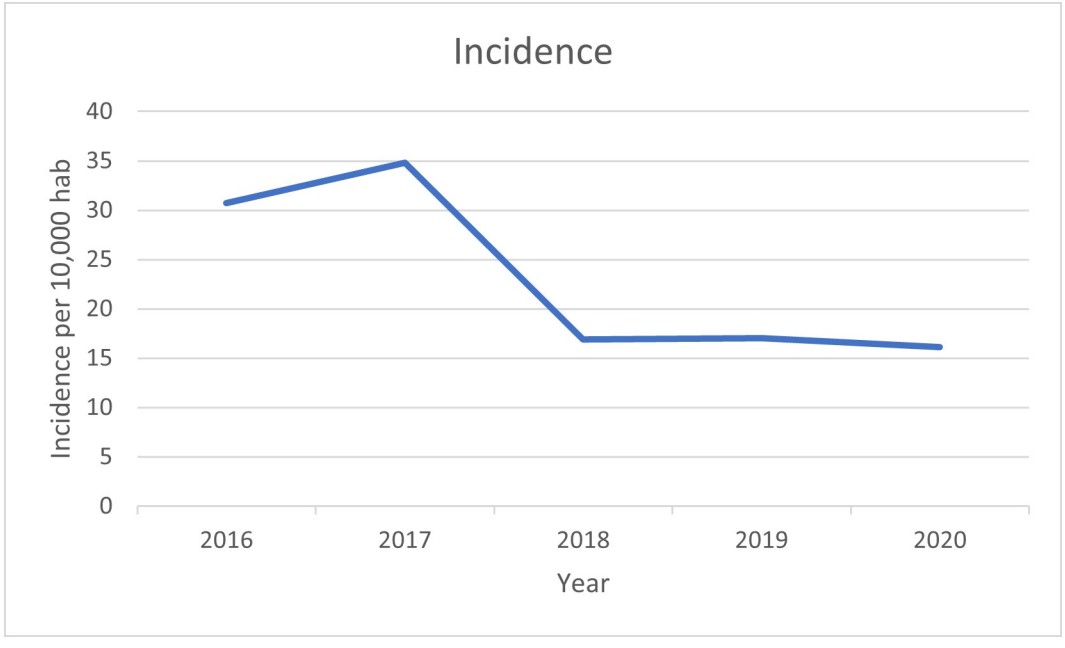

**Fig 1. Incidence per 10,000 hab.**

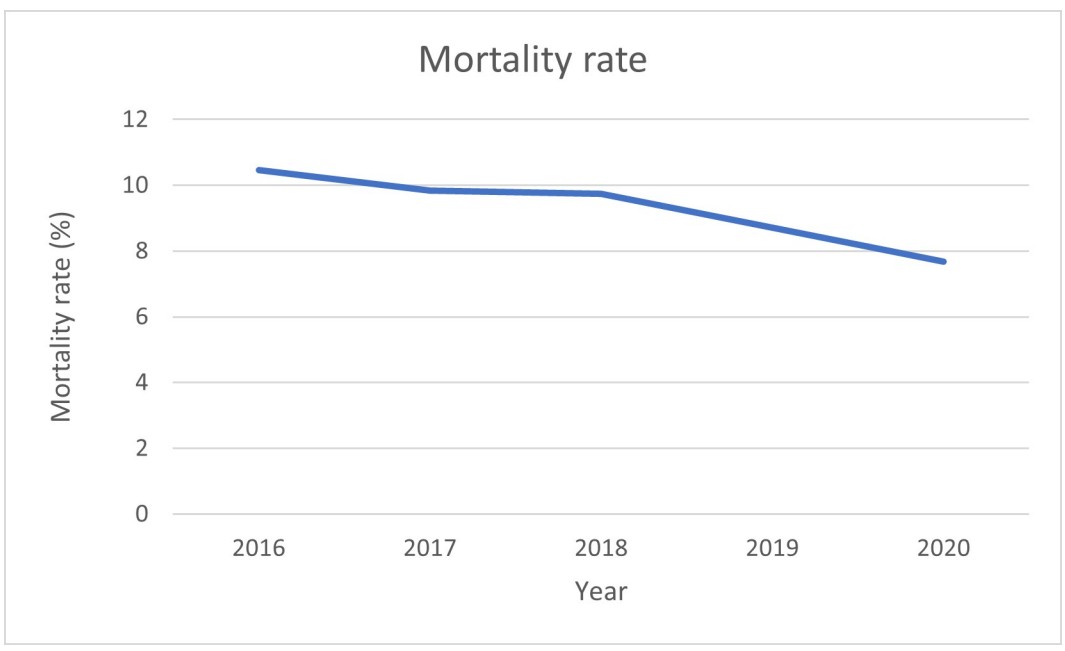

**Fig 2. Mortality rate.**

Since the introduction of PSA technique for screening, the cancer incidence and the survival rate has increased gradually each year [12]. Based on statistical modelling, diagnosed cases of advanced prostate cancer have been declining over the last decade, along with the mortality rate, which falls by an average of 3.4% each year [13]. These improvements can be attributed to early diagnosis and improved treatment options [13]. In Spain, the mortality rate

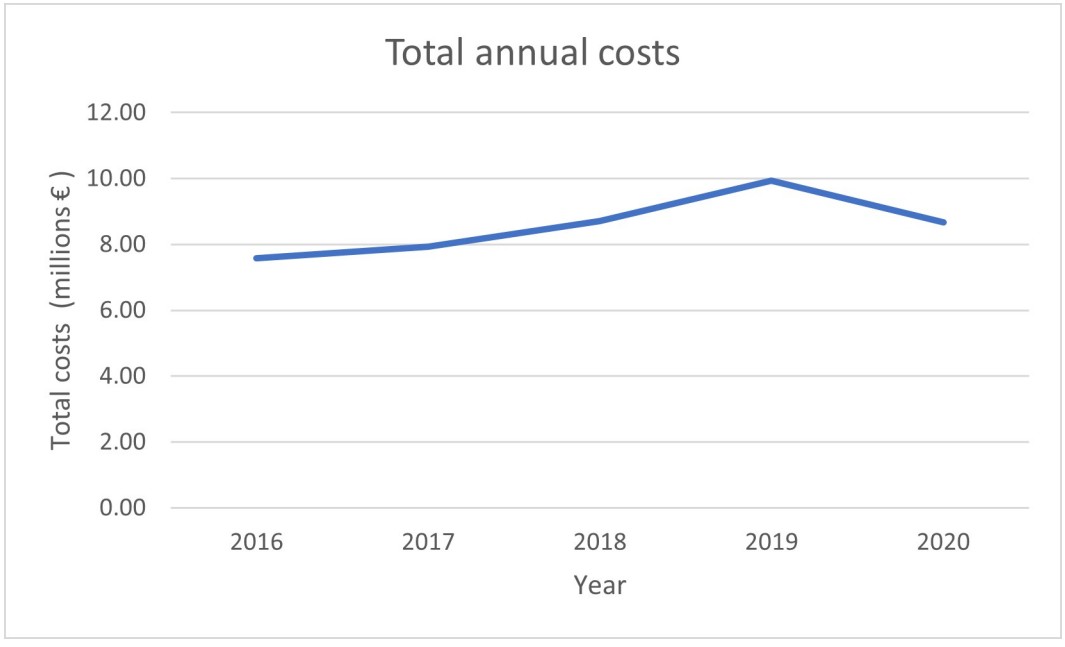

**Fig 3. Total annual costs.**

is gradually decreasing, with the highest mortality rate of 10.45% in 2016, while there is an annual mortality rate of 22.88% worldwide [14].

Although such big improvements with the introduction of PSA, there are new innovative diagnosis techniques that are currently being studied, such as the use of biomarkers [15]. This new technique allows to guide decisions on who to perform a biopsy and who to re-biopsy after negative results prior [15]. They are urgently needed to guide pre-treatment decisions [15].

Most of the cases diagnosed are localized prostate cancers, with a 20% being advanced cases or metastatic diseases [16]. The most common primary treatment for metastatic prostate was androgen deprivation therapy (ADT) for decades, but recent trends show via medical or surgical castration as the primary treatment [16].

In the research by Wahlgren, the most common CCI score found in patients with prostate cancer was 3 [17]. While 34.57% of the patients had a moderate CCI score, over 27% of the patients had a severe CCI score over 5 [18].

Regardless of survival rate, those with a comorbidity score higher than 2, had a higher mean medical cost within the first year of diagnosis, compared to those who had a comorbidity score of 2 [19]. While in Spain, the annual mean cost was 5212.98€, in the Unites States was $55,949 per person-year [20].

## Conclusion

This study provides data describing the characteristics of patients who were diagnosed with prostate cancer in Spanish hospitals over 5 years and the associated medical costs. The study showed an improvement in the quality of life of patients with the introduction of new technologies in the detection of prostate cancer. The use of these new technologies in the recent years has helped in the early detection of the disease. To improve the detection, diagnosis and treatment, further research is needed to identify new technologies in the disease management over time.

## Author Contributions

**Conceptualization:** Josep Darbà.

**Formal analysis:** Meritxell Ascanio.

**Methodology:** Meritxell Ascanio.

**Validation:** Josep Darbà, Meritxell Ascanio.

**Writing – original draft:** Meritxell Ascanio.

**Writing – review & editing:** Josep Darbà, Meritxell Ascanio.

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
