## [Decision Letter · Decision Letter 0]

8 Dec 2023

PONE-D-23-28245Prostate cancer in Spain: a retrospective database analysis of hospital incidence and the direct medical costsPLOS ONE

Dear Dr. Darbà,

Thank you for submitting your manuscript to PLOS ONE. After careful consideration, we feel that it has merit but does not fully meet PLOS ONE’s publication criteria as it currently stands. Therefore, we invite you to submit a revised version of the manuscript that addresses the points raised during the review process.

We look forward to receiving your revised manuscript.

Kind regards,

Zhaoqing Du, Ph.D

Academic Editor

PLOS ONE

Journal Requirements:

Reviewers' comments:

Reviewer's Responses to Questions

**Comments to the Author**

1. Is the manuscript technically sound, and do the data support the conclusions?

Reviewer #1: No

Reviewer #2: Yes

2. Has the statistical analysis been performed appropriately and rigorously? 

Reviewer #1: No

Reviewer #2: Yes

3. Have the authors made all data underlying the findings in their manuscript fully available?

Reviewer #1: No

Reviewer #2: Yes

4. Is the manuscript presented in an intelligible fashion and written in standard English?

Reviewer #1: Yes

Reviewer #2: Yes

5. Review Comments to the Author

Reviewer #1: The authors have conducted a good study, however, there are a lot of important and relevant information missing from the manuscript. Firstly, the authors describe the study as a multi center study but have not mentioned how many centers the data were collected from to justify their sample size. Secondly, the authors have not mentioned in detail the type of diagnostic tests, if any, that the patients were subjected to and the costs of these tests and wether or not the patients paid for these themselves. Thirdly, it will be beneficial for the authors to highlight the major treatment procedures that these patients underwent and the average cost of such procedures and wether or not the patients had multiple procedures done during the course of their treatments. The authors should also mention wether the patients benefit from any health insurance schemes if available. Overall, I believe this study can be made better and more impactful than it currently is.

Reviewer #2: The paper is solid and make a point asPSA merges as an important predictor according to this study The results are solid however the authors should clarify which software they have used for each statistical analysis.

6. PLOS authors have the option to publish the peer review history of their article (what does this mean?). If published, this will include your full peer review and any attached files.

Reviewer #1: No

Reviewer #2: No

---

## [Author Response · Author response to Decision Letter 0]

29 Jan 2024

Reviewers' comments:

Reviewer's Responses to Questions

Comments to the Author

1. Is the manuscript technically sound, and do the data support the conclusions?

Reviewer #1: No

Reviewer #2: Yes

2. Has the statistical analysis been performed appropriately and rigorously? 

Reviewer #1: No

Reviewer #2: Yes

3. Have the authors made all data underlying the findings in their manuscript fully available?

Reviewer #1: No

Reviewer #2: Yes

4. Is the manuscript presented in an intelligible fashion and written in standard English?

Reviewer #1: Yes

Reviewer #2: Yes

5. Review Comments to the Author

Reviewer #1: The authors have conducted a good study, however, there are a lot of important and relevant information missing from the manuscript. 

Firstly, the authors describe the study as a multi center study but have not mentioned how many centers the data were collected from to justify their sample size. 

The data come from a database that includes information from many centers, but we cannot provide the exact number of centers from which our data come. As indicated in the text, the database contains information from 90% of the hospitals in Spain and includes data from all regions.

Secondly, the authors have not mentioned in detail the type of diagnostic tests, if any, that the patients were subjected to and the costs of these tests and wether or not the patients paid for these themselves. 

The subjects were not included based on any diagnostic test. The patients included in the study came from a database and were selected according to its diagnosis based on IDC 10 codes. 

Thirdly, it will be beneficial for the authors to highlight the major treatment procedures that these patients underwent and the average cost of such procedures and wether or not the patients had multiple procedures done during the course of their treatments. 

The most common medical procedures have already been provided in the results section and in Table 3. Regarding costs, the database just provides the total cost of the admission, so the procedure cost is included in the total, but we can not know the exact cost of the concrete procedure. Additional information on the number of procedures performed per patient has been added.

The authors should also mention wether the patients benefit from any health insurance schemes if available. 

Information about this point has been added. In Spain, healthcare is free of charge, so the costs provided will be those paid by the national health system. 

Overall, I believe this study can be made better and more impactful than it currently is.

Reviewer #2: The paper is solid and make a point asPSA merges as an important predictor according to this study. The results are solid however the authors should clarify which software they have used for each statistical analysis.

Just Stata SE has been used to conduct the statistical analysis. This information has been updated in the methods section.

---

## [Editor Report · Decision Letter 1]

31 Jan 2024

Prostate cancer in Spain: a retrospective database analysis of hospital incidence and the direct medical costs

PONE-D-23-28245R1

Dear Dr. Josep Darbà,

We’re pleased to inform you that your manuscript has been judged scientifically suitable for publication and will be formally accepted for publication once it meets all outstanding technical requirements.

Kind regards,

Zhaoqing Du, Ph.D

Academic Editor

PLOS ONE

Additional Editor Comments (optional):

Reviewers' comments:

None.

---

## [Editor Report · Acceptance letter]

26 Feb 2024

PONE-D-23-28245R1 

PLOS ONE

Dear Dr. Darbà, 

I'm pleased to inform you that your manuscript has been deemed suitable for publication in PLOS ONE. Congratulations! Your manuscript is now being handed over to our production team.

Kind regards, 

on behalf of

Dr. Zhaoqing Du 

Academic Editor

PLOS ONE